# Anal Human Papillomavirus Infection among MSM Attending University in China: Implications for Vaccination

**DOI:** 10.3390/vaccines8020175

**Published:** 2020-04-09

**Authors:** Song Fan, Peiyang Li, Lin Ouyang, Tanwei Yuan, Hui Gong, Yi Ding, Zhenzhou Luo, Guohui Wu, Maohe Yu, Huachun Zou

**Affiliations:** 1Department of Medical Statistics, School of Public Health, Sun Yat-sen University, Guangzhou 510080, China; fansong@swmu.edu.cn (S.F.); lipy26@mail2.sysu.edu.cn (P.L.); 2Department of Social Medicine, School of Public Health, Southwest Medical University, Luzhou 646000, China; 3Department of AIDS/STD Control and Prevention, Chongqing Center for Disease Control and Prevention, Chongqing 400042, China; ouyl210@126.com (L.O.); wgh68803652@163.com (G.W.); 4School of Public Health (Shenzhen), Sun Yat-sen University, Shenzhen 510006, China; ytanwei@gmail.com; 5Department of AIDS/STD Control and Prevention, Tianjin Center for Disease Control and Prevention, Tianjin 300011, China; tjghcdc@163.com; 6Nanshan Center for Chronic Disease Control, Shenzhen 518000, China; dingeasy111@163.com (Y.D.); paulluo9909@163.com (Z.L.); 7Kirby Institute, University of New South Wales, Sydney 2052, Australia; 8Department of Community Health and Behavioral Medicine, School of Public Health, Shanghai Jiao Tong University, School of Medicine, Shanghai 200025, China

**Keywords:** vaccination, human papillomavirus, men who have sex with men, sexually transmitted infections, students

## Abstract

Men who have sex with men (MSM) attending university are a high-risk population for human papillomavirus (HPV) infection and are a neglected population of HPV vaccination programs in China. To provide evidence for HPV vaccination policies, we conducted this study to examine the prevalence and factors associated with anal HPV infection among MSM attending university in China. A self-administered online questionnaire was conducted to collect information on social demographics and sexual behaviors. A self-collected rectal swab specimen was collected to test for 37 HPV types. A total of 426 participants were tested for HPV. The median age was 20 years. HPV prevalence was 37.5% for any type, 29.8% for nine-valent vaccine types, 24.6% for four-valent vaccine types, 11.5% for HPV-16/18, and 15.7% for HPV-6/11. Men enrolled in a technical diploma, living in Northern China, having more than two sex partners, being bottom or versatile in anal sex, and having a human immunodeficiency virus (HIV) testing history were more likely to have positive anal HPV of any type. Our study found a high prevalence of anal HPV infection among MSM attending university in China, with HPV vaccine-preventable types being the most popular types in this group. Thus, our findings highlight the urgency of promoting HPV vaccination among teenage MSM.

## 1. Introduction

Human papillomavirus (HPV) has been identified as one of the most common sexually transmitted infections (STIs) worldwide [1]. Most HPV infections appear to be transient and asymptomatic, but persistent infections can cause severe malignancies [2,3], globally accounting for 90% of genital warts [4], 88% of anal cancers [5], 50% of penile cancers [5], and 31% of oral and pharyngeal cancers [5]. HPV and human immunodeficiency virus (HIV) could have a synergistic effect. Specifically, HPV infection, which may facilitate the invasion of pathogens, is a significant risk factor (3.5-times increased risk) for HIV acquisition [6]. Similarly, HIV infection can increase the risk of HPV infection and the failure of HPV clearance through weakening the immune function of the body [7].

Although there is an abundance of literature on HPV among men who have sex with men (MSM), data on HPV among very young MSM are still lacking. A systematic review that only included 698 MSM ≤ 25 years of age, all of whom were from North America and Europe, found that the rate of anal HPV-16 infection was up to 16% among MSM aged ≤ 20 years and up to 18% among those aged 21–25 years [8]. There is no published literature that focuses on HPV prevalence among young and teenage MSM in Asia. MSM attending university in China are a subgroup in the MSM community facing high risks of HIV and other STIs [9]. A systematic review found that the prevalence rates of HIV and syphilis among this population were 4.1% and 4.7%, respectively [10]. It was estimated that there were approximately 1.7 million MSM attending university in China aged 18–22 years in 2017 [11]. This group, which has a high prevalence of unprotected anal intercourse (UAI) and low condom use, are facing increasing risks of HIV and STIs [9]. Consistent with HIV infection, these behaviors may also put MSM attending university at potential risk of HPV transmission [7].

MSM are a neglected population in HPV prevention programs around the world. Three prophylactic HPV vaccines, directed against certain HPV types, are currently available for the prevention of HPV-related disease: the bivalent vaccine is recommended to females, and the quadrivalent and the nine-valent vaccines are recommended to both males and females [12]. However, current recommendations of HPV vaccination worldwide mainly target women, and very few countries provide HPV vaccination for men [13]. Despite the unavailability of a male HPV vaccination program, researchers have argued for targeted HPV vaccinations among teenage MSM in China [14]. Until that day comes, concrete evidence is needed. To provide evidence to support the promotion of HPV vaccination programs in young MSM in China, we conducted this study to characterize HPV infections and the associated risk factors among MSM attending university in China.

## 2. Materials and Methods

### 2.1. Study Population

The methodology of the University Student HIV Test Intervention Study (the UniTest Study) has previously been published elsewhere [15]. Briefly, MSM attending university were recruited in four regions (Chongqing in Western China, Guangdong in Southern China, Shandong in Eastern China, and Tianjin in Northern China) from January to April in 2019 via advertising on the Internet, social network applications, peer referrals, and clinics. Participants were eligible if they were male, aged 16 years or older, university students, and self-reported to have sex with men. At enrollment, each participant provided informed consent. This study was conducted with the approval of the institutional review boards of Sun Yat-sen University (SYSU-SPH2018044).

### 2.2. Data Collection

Participants were required to finish a self-administrated online questionnaire to collect information regarding sociodemographics, sexual behavioral, HIV testing history, and alcohol and recreational drug use.

### 2.3. Sample Collection and Testing

Participants conducted a self-collected anal swab sample for HPV testing. The polymerase chain reaction (PCR) HPV genotyping test (Hybribio Biochemical Co., Ltd., Chaozhou, China) was used to determine the HPV-type distribution in specimens. The PCR test amplified the target HPV DNA for 37 anogenital HPV types, which included HPV-6, 11, 16, 18, 26, 31, 33, 34, 35, 39, 40, 42, 43, 44, 45, 51, 52, 53, 54, 55, 56, 57, 58, 59 61, 66, 67, 68, 69, 70, 71, 72, 73, 81, 82, 83, and 84.

HIV testing was conducted by trained doctors with rapid HIV testing (Alere^®^ Determine HIV-1/2, Alere Medical Co., Ltd., Matsudo, Japan). Participants that tested positive for HIV were referred to local centers for disease control and prevention (CDC) for confirmation testing and subsequent services.

### 2.4. Variables and Definitions

Sociodemographic variables of interest included age, education, academic performance (based on the self-reported grade point average (GPA)), dwelling location, sexual orientation, and the gender of sex partners. Sexual behaviors included age of first anal intercourse, number of sexual partners in the past 3 months, commercial sex history, compulsive sex history, engaging in group sex in the past 3 months, role of anal sex in the past 3 months, unprotected anal sex in the past 3 months, and condom use during last anal sex encounter.

The HPV detection variables consisted of type-specific HPV prevalence, any HPV type, high-risk HPV types, low-risk HPV types, single infection types, multiple infection types [16], and vaccine-preventable HPV types. Four-valent HPV types included HPV-6, 11, 16, and 18. Nine-valent HPV types included HPV-6, 11, 16, 18, 31, 33, 45, 52, and 58. High-risk HPV types included HPV-16, 18, 31, 33, 35, 39, 45, 51, 52, 56, 58, 59, and 68. Low-risk HPV types included HPV-6, 11, 26, 34, 40, 42, 43, 44, 53, 54, 55, 57, 61, 66, 67, 69, 70, 71, 72, 73, 81, 82, 83, and 84 [17].

### 2.5. Statistical Analyses

Descriptive statistics included summary measures of the mean or median and interquartile range (IQR) for continuous variables (e.g., age) and frequency tables for categorical variables (e.g., positive HPV detections). HPV prevalence was given as a percentage and corresponding 95% confidence intervals (CIs) for binomial proportions. Adjusted odds ratios (AORs) and 95% CIs of HPV infections were calculated according to characteristics using multivariate logistic regression models. Predictors with *p* < 0.10 in the univariate analyses were included in the multiple logistic regression model to determine factors associated with HPV infections. The chi-square test with trends was used to demonstrate the significance of the trends in HPV prevalence with the time since first anal sex. Analyses were conducted using STATA MP 16 (StataCorp LLC, College Station, TX, USA).

## 3. Results

### 3.1. Participant Characteristics

A total of 447 MSM attending university were enrolled in the study from January to April 2019. The demographic characteristics of the participants are shown in Table 1. Ages ranged from 17 to 27 years, with a median of 20 (IQR 19–21) years. Most were undergraduate students (342, 80.3%), self-reported above-average academic performances (338, 79.3%), and of homosexual orientation (364, 85.4%). Twelve participants (2.8%) were HIV-positive.

### 3.2. Sexual and High-Risk Behaviors

Table 1 also shows the sexual behaviors among MSM attending university by any HPV detection. Most of the participants (155, 96.9%) reported having sex with men only. A significantly higher percentage of participants with positive HPV diagnosis reported that the age of first anal intercourse (AFAI) with men was less than 18 years old (62.5% vs. 50.8%; *p* = 0.018) and having > 2 sex partners (25.0% vs. 14.3%; *p* = 0.012). Participants with a positive HPV diagnosis practiced significantly more receptive anal sex (45.6% vs. 23.9%; *p* < 0.001). Compared to HPV-negative participants, HPV-positive participants had a greater history of HIV testing (94.4% vs. 85.7%, *p* = 0.006). About one in five of all participants (83, 19.5%) had sex after drinking alcohol. Approximately one-third of all participants (126, 29.6%) reported recreational drug use.

### 3.3. Anal HPV Detection and Types

A total of 431 anal swabs were collected and tested for HPV types—Of which, five failed to test, with unsatisfactory samplings. Table 2 shows the prevalence of anal HPV types among MSM attending university. The prevalence of anal HPV infections of any, high-risk, low-risk, and multiple types were 37.5%, 25.1%, 26.3%, and 19.0%, respectively. HPV-6 (8.9%), HPV-16 (8.7%), HPV-11 (7.5%), and HPV-18 (4.0%) were the most prevalent HPV types. Of those who were infected with any type of HPV, 79.4% (127/160) were infected with at least one type of nine-valent vaccine-preventable HPV type. The individual level of each HPV type is listed in Appendix A.

### 3.4. Risk Factors of Anal HPV Infection

We found positive trends between positive anal HPV infections and time since first anal sex (Figure 1). The proportion of participants with anal HPV of any type, any high-risk type, any two-valent (HPV-16/18) vaccine-preventable type, any four-valent vaccine-preventable type, and any nine-valent vaccine-preventable type increased significantly with time since the first anal sex (all *p* < 0.05).

Figure 2 shows the factors associated with anal HPV infections among MSM attending university. In a multivariate analysis, participants enrolled in technical diploma programs (AOR: 1.88; 95% CI: 1.11–3.20), living in northern China (AOR: 1.98; 95% CI: 1.10–3.56), having more than two sexual partners (AOR: 2.29; 95% CI: 1.18–4.46), playing a receptive role (AOR: 7.16; 95% CI: 2.98–17.18) or a versatile role (AOR: 2.93; 95% CI: 1.29–6.66) in anal sex, and having an HIV testing history (AOR: 2.44; 95% CI: 1.11–5.36) were more likely to be infected with anal HPV of any type.

## 4. Discussion

We found a high HPV prevalence among MSM attending university in China. The great majority of infected HPV types could have been prevented by HPV vaccination. To our knowledge, this is the first study to characterize the prevalence and types of anal HPV infections in this young population in China.

Our study found that the prevalence of any anal HPV infection among MSM attending university was 37.5%, which was lower than the overall prevalence of 52–66% among MSM with HIV-negative status in the general population in China [18,19]. The possible reasons are as follows: compared with general MSM, MSM attending university are younger and have fewer sexual experiences, so they may be less exposed. In addition, they are generally more educated and may have a stronger sense of self-protection [10]. We found similar prevalence rates for HPV 6/11, 16/18, and 16 to that among MSM aged 20–29 years in another study that included 536 MSM in southern China [20].

According to our results, the majority of participants were infected with HPV vaccine-preventable types, indicating that promoting HPV vaccinations could potentially be an effective way to prevent HPV infections among this young population. In our study, HPV-16 and 18 were the most prevalent high-risk HPV types. Studies have found that the persistence of high-risk HPV types, specifically HPV-16/18, is an important risk factor for the development of anal cancer [21]. Fortunately, both four-valent and nine-valent HPV vaccines can protect MSM from these two HPV types. HPV-6 and 11 types, which can cause genital warts [4,22], were the most prevalent low-risk HPV types among MSM attending university, and these two types could be protected by both the four-valent and nine-valent HPV vaccines. A previous study found that HPV vaccinations in males were highly effective in largely preventing anogenital warts and anal cancers [23]. The modeling analysis predicted that a vaccination program for males only with a certain coverage (80%) could achieve a reduction of 90% in HPV infections in MSM. In addition, the study also found targeted vaccination programs that commenced after sexual activity were much less effective, [24] suggesting that vaccinating males before their sex debut might substantially reduce the burden of HPV-related diseases.

Our study also found some risk factors associated with anal HPV infections. The proportion of anal HPV infections increased with time since the first anal intercourse among MSM attending university. With the increase of sexual activity, they are more likely to be exposed to HPV, which correspondingly increases the risk of HPV infection. Similar to the results in the general MSM population, playing a receptive role in anal sex and having multiple sex partners are also risk factors of anal HPV infections among MSM attending university [25,26]. Furthermore, the use of geosocial networking gay applications (gay apps) is associated with risk behaviors, such as casual sex, group sex, drug use, etc., which lead to an increased risk of infection with HIV and HPV [27].

We found that participants enrolled in technical diploma programs were more likely to be infected with HPV than undergraduates. In the Chinese education system, the admission scores and learning tasks of technical diploma programs are lower than those of undergraduates [28]. Students in technical schools may have less academic burdens and have more free time than undergraduates, so they have comparatively more time to engage in sexual activities. Further studies may be needed to explain this result. Participants from the north had a higher proportion of HPV infections than those from the south, suggesting that there are regional differences in HPV infections. Similarly, in previous studies among the general MSM population in China, the proportions of anal HPV infections in northern cities (e.g., 71% for Beijing [29] and 62% for Tianjin [30]) were higher than those in southern cities (e.g., 48% Guangzhou [26] and 36% for Shenzhen [31]). The differences in HPV infections may be related to northern and southern Chinese living habits. In the north, group bathing in the bathhouse is preferred, while in the south, individual bathing at home is more common [32]. MSM who live in the north are more likely to engage in high-risk behaviors in the bathhouse, leading to a higher risk of HPV infections than those living in the south. We also found participants with a history of HIV testing had a higher prevalence of HPV infections. Previous studies found that serosorting may lead to reduced condom uses [33]. However, mathematical modeling suggested that inappropriate serosorting may increase the risk of STI acquisitions [34].

Our study had limitations. Firstly, this study was cross-sectional, thus limiting our ability to differentiate between transient HPV infections and persistent HPV infections. HPV detection at a single time cannot represent persistent HPV infections. A prospective study found that 66% and 90% of HPV infections would be cleared within 12 and 24 months, respectively, in men [35]. Consecutive HPV testing will be conducted at the 12-month visit to further understand the natural history of HPV infections among MSM attending university. Secondly, participants conducted self-operated anal samplings for HPV detection. Compared with professional-operated sampling, self-operated sampling may increase sampling failure but with the same accuracy [36]. The lessons learned will guide the implementation of home-based self-sampling for HPV and other STI testing in the future. Thirdly, the questionnaires were self-reported, and hence, recall bias might have occurred. Fourthly, we used only anal swab samples to test for HPV in cells, not anal biopsy samples to test for HPV in tissues. Swab sampling for detecting viruses is likely to sample HPV DNA present in the adjacent tissue or present only on the surface of the lesion, which may not necessarily be linked to the proliferation itself [16]. An existing systematic review found there was no difference in the HPV-16 detection rate between biopsies and cells, whereas the detection rate of other high-risk HPV types, including HPV-35, 51, and 68, was five times lower in biopsies than in cells [37]. However, biopsies can help diagnose precancerous lesions and could be considered in future studies. Finally, the method we used for HPV testing had high sensitivity and specificity in cervical samples [38], with a limit of detection of 500 IU/mL (international units per milliliter) [39]. Nevertheless, little is known about the consistency of this method in anal samples among MSM, indicating the need for further studies.

## 5. Conclusions

Our study found a high prevalence of anal HPV infections among MSM attending university, especially for HPV vaccine-preventable types, suggesting that this population are a potential target group for HPV vaccinations. Prospective studies are needed to understand the natural history and influencing factors of HPV infections in this young population and to evaluate the effectiveness of HPV vaccinations in young males in China.

## Figures and Tables

**Figure 1 vaccines-08-00175-f001:**
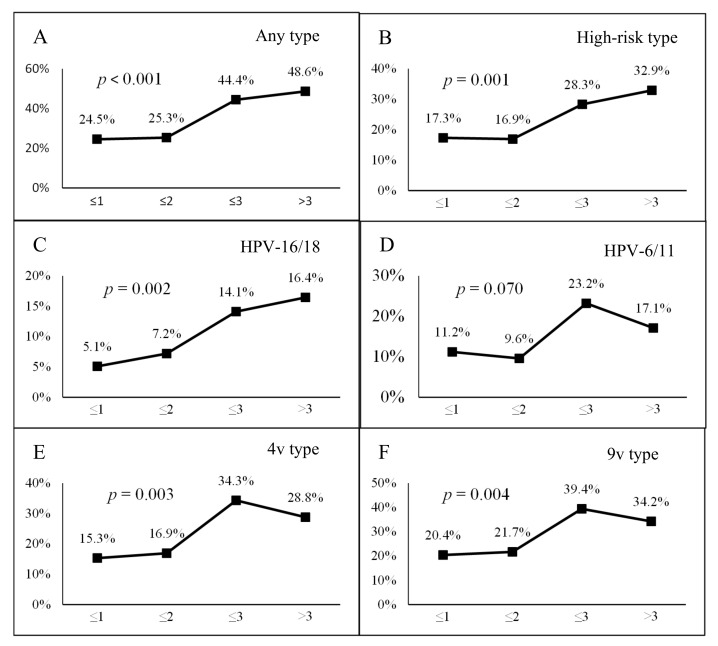
Proportions of any anal human papillomavirus (HPV) detection and the time since the first anal intercourse among men who have sex with men (MSM) attending university in China. The proportion of men with anal HPV of (**A**) any type, (**B**) high-risk type, (**C**) HPV-16 or 18 types, (**D**) HPV-6 or 11 types, (**E**) four-valent (HPV-6,11,16, and 18) types, and (**F**) nine-valent (HPV-6, 11, 16, 18, 31, 33, 45, 52, and 58) types with years since the first anal intercourse.

**Figure 2 vaccines-08-00175-f002:**
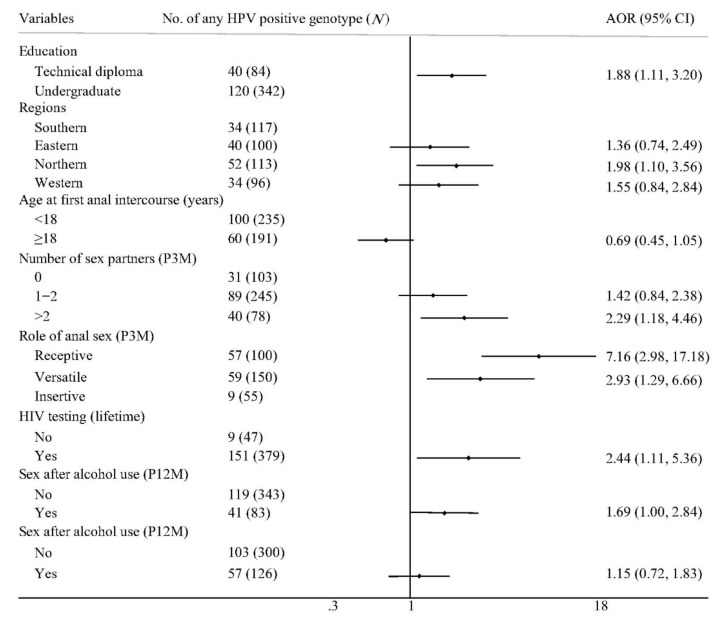
Factors associated with any anal HPV detection among MSM attending university in China. MSM: men who have sex with men, HPV: human papillomavirus, AOR: adjusted odds ratio, CI: confidence interval, P3M: past 3 months, and P12M: past 12 months. The symbol 
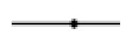
 represents the AOR value and its 95% CI, the dot represents the AOR value, and the length of the line represents the 95% CI range.

**Table 1 vaccines-08-00175-t001:** Demographic characteristics and sexual behaviors among MSM attending university in China (*n* = 426).

Variables	*n*	Any HPV-Positive (*n* = 160)		Any HPV-Negative (*n* = 266)		*p*
		No.	%	No.	%	
Age (years)						
≤20	225	78	48.8	147	55.3	0.192
>20	201	82	51.3	119	44.7	
Education						
Technical diploma	84	40	25.0	44	16.5	0.034
Undergraduate	342	120	75.0	222	83.5	
Academic performance						
Average and below	88	33	20.6	55	20.7	0.990
Above average	338	127	79.4	211	79.3	
Regions						
Southern	117	34	21.3	83	31.2	0.058
Eastern	100	40	25.0	60	22.6	
Northern	113	52	32.5	61	22.9	
Western	96	34	21.3	62	23.3	
Sexual orientation						
Other	20	6	3.8	14	5.3	0.106
Bisexual	42	10	6.3	32	12.0	
Homosexual	364	144	90.0	220	82.7	
Gender of sex partners (*n* = 414)						
Only male	401	155	98.1	246	96.1	0.261
Male and female	13	3	1.9	10	3.9	
Age at first anal intercourse (years)						
<18	235	100	62.5	135	50.8	0.018
≥18	191	60	37.5	131	49.2	
Number of sex partners (P3M)						
0	103	31	19.4	72	27.1	0.012
1–2	245	89	55.6	156	58.6	
>2	78	40	25.0	38	14.3	
Group sex (P3M)						
No	349	129	80.6	220	82.7	0.589
Yes	77	31	19.4	46	17.3	
Role in anal sex (P3M) (*n* = 305)						
Receptive	100	57	45.6	43	23.9	<0.001
Versatile	150	59	47.2	91	50.6	
Insertive	55	9	7.2	46	25.6	
Unprotected anal intercourse (P3M)						
No	291	103	64.4	188	70.7	0.176
Yes	135	57	35.6	78	29.3	
HIV testing (lifetime)						
No	47	9	5.6	38	14.3	0.006
Yes	379	151	94.4	228	85.7	
HIV status of sex partners (P3M) (*n* = 196)						
Negative	144	51	64.6	93	79.5	0.065
Positive	5	3	3.8	2	1.7	
Unknown	47	25	31.6	22	18.8	
Sex after alcohol use (P12M)						
No	343	119	74.4	224	84.2	0.013
Yes	83	41	25.6	42	15.8	
Recreational drug used in sex (lifetime)						
No	300	103	64.4	197	74.1	0.034
Yes	126	57	35.6	69	25.9	
HIV status						
Negative	414	151	94.4	263	98.9	0.012
Positive	12	9	5.6	3	1.1	

P3M: past 3 months, P12M: past 12 months, MSM: men who have sex with men, HPV: human papillomavirus, and HIV: human immunodeficiency virus.

**Table 2 vaccines-08-00175-t002:** Type-specific anal HPV detection among MSM attending university in China (*n* = 426).

Type of Infection	Positive Cases	Prevalence (95% CI) (%)
Any HPV type	160	37.5 (32.9, 42.2)
Any high-risk type	107	25.1 (21.0, 29.3)
Any low-risk type	112	26.3 (22.1, 30.5)
No. of HPV genotype detections		
Median and range	Median: 2	Min: 1, Max: 9
1 type	79	18.5 (14.8, 22.3)
2 types	44	10.3 (7.4, 13.2)
3 types	18	4.2 (2.3, 6.1)
4+ types	19	4.5 (2.5, 6.4)
Single infection		
Single high-risk types	35	8.2 (5.6, 10.8)
Single low-risk types	44	10.3 (7.4, 13.2)
Mixed infection		
Multiple high-risk types	13	3.1 (1.4, 4.7)
Multiple low-risk types	9	2.1 (0.7, 3.5)
Both high-risk and low-risk types	59	13.8 (10.6, 17.1)
High-risk type		
HPV 16	37	8.7 (6.0, 11.4)
HPV 18	17	4.0 (2.1, 5.9)
HPV 31	5	1.2 (0.1, 2.2)
HPV 33	7	1.6 (0.4, 2.9)
HPV 35	2	0.5 (−0.2, 1.1)
HPV 39	11	2.6 (1.1, 4.1)
HPV 45	9	2.1 (0.7, 3.5)
HPV 51	16	3.8 (1.9, 5.6)
HPV 52	16	3.8 (1.9, 5.6)
HPV 56	8	1.9 (0.6, 3.2)
HPV 58	9	2.1 (0.7, 3.5)
HPV 59	1	0.2 (−0.2, 0.7)
HPV 68	17	4.0 (2.1, 5.9)
Low-risk type		
HPV 6	38	8.9 (6.2, 11.6)
HPV 11	32	7.5 (5.0, 10.0)
HPV 40	14	3.3 (1.6, 5.0)
HPV 42	8	1.9 (0.6, 3.2)
HPV 43	5	1.2 (0.1, 2.2)
HPV 44	3	0.7 (−0.1, 1.5)
HPV 53	7	1.6 (0.4, 2.9)
HPV 54	6	1.4 (0.3, 2.5)
HPV 55	3	0.7 (−0.1, 1.5)
HPV 61	10	2.3 (0.9, 3.8)
HPV 66	3	0.7 (−0.1, 1.5)
HPV 67	10	2.3 (0.9, 3.8)
HPV 69	1	0.2 (−0.2, 0.7)
HPV 70	1	0.2 (−0.2, 0.7)
HPV 71	1	0.2 (−0.2, 0.7)
HPV 72	2	0.5 (−0.2, 1.1)
HPV 73	5	1.2 (0.1, 2.2)
HPV 81	8	1.9 (0.6, 3.2)
HPV 82	5	1.2 (0.1, 2.2)
HPV 84	6	1.4 (0.3, 2.5)
Vaccine-preventable types		
9vHPV	127	29.8 (25.5, 34.2)
4vHPV	105	24.6 (20.5, 28.8)
HPV 6/11	67	15.7 (12.3, 19.2)
HPV 16/18	49	11.5 (8.5, 14.5)
No. of vaccine-preventable type detections		
Median and range	Median: 1	Min: 1, Max: 5
1 type	95	22.3 (18.3, 26.3)
2 types	25	5.9 (3.6, 8.1)
3 types	4	0.9 (0, 1.9)
4+ types	3	0.7 (−0.1, 1.5)

Note: HPV-26, 34, 57, and 83 were not detected. MSM: men who have sex with men; CI: confidence interval; HPV: human papillomavirus; HPV-6/11: HPV-6 or 11; HPV-16/18: HPV-16 or 18; 4vHPV: 4-valent HPV types, including HPV-6, 11, 16, and 18; and 9vHPV: 9-valent HPV types, including HPV-6, 11, 16, 18, 31, 33, 45, 52, and 58. High-risk HPV types include HPV-16, 18, 31, 33, 35, 39, 45, 51, 52, 56, 58, 59, and 68. Low-risk HPV types include HPV-6, 11, 26, 34, 40, 42, 43, 44, 53, 54, 55, 57, 61, 66, 67, 69, 70, 71, 72, 73, 81, 82, 83, and 84. Any type: any of the 37 HPV types mentioned above.

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
