# Peer review of "Anal Human Papillomavirus Infection among MSM Attending University in China: Implications for Vaccination"

_vaccines, 2020, doi:10.3390/vaccines8020175_

Round 1

Reviewer 1 Report

Authors have presented a manuscript entitled ´Anal human papillomavirus infection among MSM attending university in China: implications for vaccination ´ to be considered for publication in the journal Vaccines.

The article has a very timely and important topic regarding gender-neutral HPV vaccination and although the results are not totally novel they are interesting, particularly the risk associations which could be more clearly emphasized.

Moreover, the following revisions are needed before the article can be accepted for publication.

  1. A more detailed dissemination regarding the limitation of the genotyping method eg. sensitivity, if any.
  2. Clear discussion on the limitations in using self-administered swab samples compared with anal biopsies in order to understand prevalence of active HPV infections.
  3. Authors should analyze the multiple HPVs infection patterns (half of the data) as well.
  4. Important to clarify, how multiple HPVs infections were integrated or not (and why) in the type level prevalence distribution calculations. If the multiple infections were included, how were they weighted. See example eg. from Clin Microbiol Infect. 2015;21(6):605.e11–605.e6.05E19. doi:10.1016/j.cmi.2015.02.009.
  5. The raw individual-level but anonymized prevalence data should be available as a supplementary data.
  6. Most recent literature on anal swab and biopsy HPV prevalence should be cited.
  7. The English of the manuscript could be revised.

Author Response

Respond to Reviewer 1

Comments and Suggestions for Authors

Authors have presented a manuscript entitled ´Anal human papillomavirus infection among MSM attending university in China: implications for vaccination ´ to be considered for publication in the journal Vaccines.

The article has a very timely and important topic regarding gender-neutral HPV vaccination and although the results are not totally novel they are interesting, particularly the risk associations which could be more clearly emphasized.

Moreover, the following revisions are needed before the article can be accepted for publication.

  1. A more detailed dissem]ination regarding the limitation of the genotyping method eg. sensitivity, if any.

Author response 1: Thanks for your suggestions. We added the following wording “Finally, the method we used for HPV testing had high sensitivity and specificity in cervical samples. Nevertheless, little is known about the consistency of the method in anal samples among MSM, indicating further study.” in discussion (Line 238-240).

  1. Clear discussion on the limitations in using self-administered swab samples compared with anal biopsies in order to understand prevalence of active HPV infections.

Author response 2: We added the following wording “Fourthly, we used only anal swab samples to test for HPV in cells, not anal biopsy samples to test for HPV in tissues. Swab sampling for detecting viruses is likely to sample HPV DNA present in the adjacent tissue or present only on the surface of the lesion, which may not necessarily be linked to the proliferation itself. The existing systematic review found there was no difference in HPV-16 detection rate between biopsies and cells, whereas the detection rate of other high-risk HPV types including HPV-35, 51 and 68, was five times lower in biopsies than in cells. However, biopsies can help diagnose precancerous lesions and could be considered in future studies.” in discussion (Line 231-238).

  1. Authors should analyze the multiple HPVs infection patterns (half of the data) as well.

Author response 3: We added the multiple HPVs infection patterns results in Table 2.

  1. Important to clarify, how multiple HPVs infections were integrated or not (and why) in the type level prevalence distribution calculations. If the multiple infections were included, how were they weighted. See example eg. from Clin Microbiol Infect. 2015;21(6):605.e11–605.e6.05E19. doi:10.1016/j.cmi.2015.02.009.

Author response 4: Thanks for your suggestions. We have learned the methods of this document and cited them in methods.

  1. The raw individual-level but anonymized prevalence data should be available as a supplementary data.

Author response 5: We added the raw individual-level HPV prevalence data as a supplementary file.

  1. Most recent literature on anal swab and biopsy HPV prevalence should be cited.

Author response 6: We have updated the citations accordingly.

  1. The English of the manuscript could be revised.

Author response 7: Thanks for your suggestions. We duly revised the English of the manuscript.

Reviewer 2 Report

This is a well written, clear and timely piece. This manuscript consolidates existing scientific information on HPV infections among MSM, but, also addresses current information gaps on HPV prevalence and socio-demographic risk factors among young, Asian MSM in university settings.

  1. Manuscript will require minor revision in text editing. 
  2.  Authors can use phrases like “participants with positive HPV diagnosis rather than use HPV status as an adjective to describe participants i.e. HPV positive participants.

Discussion/conclusion: This section looks like it was hastily done and will require the most amount of revision. Significant findings in the result section ought to be adequately summarized, discussed and compared to other similar studies. Provide citations from peer-reviewed scientific journals as suggested by reviewer

Author Response

Respond to Reviewer 2

Comments and Suggestions for Authors

This is a well written, clear and timely piece. This manuscript consolidates existing scientific information on HPV infections among MSM, but, also addresses current information gaps on HPV prevalence and socio-demographic risk factors among young, Asian MSM in university settings.

  1. Manuscript will require minor revision in text editing. 

Author response: Thank you for your detailed suggestions on revision. We revised each issue you marked in the review document. Please review the revised manuscript.

Supplementary note:

  • Questions about the recommended population for the available HPV vaccine.

Author response: Thank you for your suggestions. In the revised manuscript, we have added the content about the bivalent vaccine recommended only for females, and the quadrivalent and nine-valent recommendations for both males and females.

  • Did this sample include both HIV-positive and HIV-negative MSM?

Author response: Yes, our sample included 12 HIV-positive MSM. Please see Table 1 ‘HIV status’.

  • Line 174: You probably meant “having no HIV testing history”?

Author response: We found participants with a history of HIV testing had a higher prevalence of HPV infection. Previous studies found serosorting may lead to reduced condom use. athematical modeling suggested that inappropriate serosorting may increase the risk of STI acquisition.

  1. Authors can use phrases like “participants with positive HPV diagnosis rather than use HPV status as an adjective to describe participants i.e. HPV positive participants.

Author response: Thank you for your suggestion. We have updated the relevant expressions.

Discussion/conclusion: This section looks like it was hastily done and will require the most amount of revision. Significant findings in the result section ought to be adequately summarized, discussed and compared to other similar studies. Provide citations from peer-reviewed scientific journals as suggested by reviewer

Author response: Thank you for your good suggestion. We have reorganized Discussion so it reads more smoothly. We also added literature citations where applicable.

Round 2

Reviewer 1 Report

Authors have sufficiently responded most of the reviewers questions. However, regarding the previous review question below, two clarifications should still be added 1) explaining the difference in detecting environmental HPV exposure (transient HPVs in anogenital regions) and progressive anal HPV infection in this study. Moreover, as authors claim the genotyping method having high sensitivity. Hence, the authors should address 2) what is the detection limit with the used genotyping method, as such, an estimation of what amount of HPV type DNA or how many copies of the HPV type genome the sample should have to be detected with the method used.

  1. A more detailed dissemination regarding the limitation of the genotyping method eg. sensitivity, if any.

Author response 1: Thanks for your suggestions. We added the following wording “Finally, the method we used for HPV testing had high sensitivity and specificity in cervical samples. Nevertheless, little is known about the consistency of the method in anal samples among MSM, indicating further study.” in discussion (Line 238-240).
